# The Dynamics of Tumor-Infiltrating Myeloid Cell Activation and the Cytokine Expression Profile in a Glioma Resection Site during the Post-Surgical Period in Mice

**DOI:** 10.3390/brainsci12070893

**Published:** 2022-07-07

**Authors:** Jescelica Ortiz-Rivera, Alejandro Albors, Yuriy Kucheryavykh, Jeffrey K. Harrison, Lilia Kucheryavykh

**Affiliations:** 1Department of Biochemistry, School of Medicine, Universidad Central de Caribe, Bayamon, PR 00956, USA; ale.albors@gmail.com (A.A.); yuriy.kucheryavykh@uccaribe.edu (Y.K.); lilia.kucheryavykh@uccaribe.edu (L.K.); 2Department of Pharmacology and Therapeutics, College of Medicine, University of Florida, Gainesville, FL 32610, USA; jharriso@ufl.edu

**Keywords:** glioma, tumor resection, microglia, cytokines

## Abstract

Glioblastoma is the most aggressive brain cancer and is highly infiltrated with cells of myeloid lineage (TIM) that support tumor growth and invasion. Tumor resection is the primary treatment for glioblastoma; however, the activation state of TIM at the site of tumor resection and its impact on glioma regrowth are poorly understood. Using the C57BL/6/GL261 mouse glioma implantation model, we investigated the state of TIM in the tumor resection area during the post-surgical period. TIM isolated from brain tissue at the resection site were analyzed at 0, 1, 4, 7, 14, and 21 days after tumor resection. An increase in expression of CD86 during the first 7 days after surgical resection and then upregulation of arginase 1 from the 14th to 21st days after resection were detected. Cytokine expression analysis combined with qRT-PCR revealed sustained upregulation of IL4, IL5, IL10, IL12, IL17, vascular endothelial growth factor (VEGF), and monocyte chemoattractant protein 1 (MCP1/CCL2) in TIM purified from regrown tumors compared with primary implanted tumors. Flow cytometry analysis revealed increased CD86^+^/CD206^+^ population in regrown tumors compared with primary implanted tumors. Overall, we found that TIM in primary implanted tumors and tumors regrown after resection exhibited different phenotypes and cytokine expression patterns.

## 1. Introduction

Glioblastoma (GBM) is the most aggressive and deadly type of brain cancer and is characterized by an extremely poor disease progression prognosis. The median survival of glioma patients is 9–12 months after diagnosis using standard treatment protocols, such as tumor resection, followed by chemo- and radiation therapy [1]. Glioma’s resistance to radiation and chemotherapy and the high probability of recurrence after surgical resection represent a major challenge in current GBM treatment.

Tumor resection is the first-line therapy for GBM; however, the invasive nature of GBM cells reduces the efficacy of surgical resection, and a majority of GBMs recur shortly afterwards. Single glioma cells left after surgical resection in the tumor surrounding healthy brain parenchyma soon lead to tumor recurrence. The effects of radiation and chemotherapy are temporary, in large part due to the ability of glioma cells to restrict the absorption of therapeutic chemicals [2,3], the increased activity of anti-apoptotic signaling [4,5], and the supportive role of the tumor microenvironment, with microglia playing a critical part [6,7]. Despite the known role of tumor-infiltrating microglia on GBM progression, the activation of microglia at the tumor resection site and its contribution to tumor regrowth have not been well-investigated. Consequently, the supportive role of microglia on glioma tumor relapse is not taken into consideration in current treatment protocols.

Myeloid-like cell populations, also known as tumor-infiltrating macrophages (TIMs), which encompass both brain microglia and bone marrow/peripheral-derived cells, contribute up to 30% of brain tumor mass and facilitate tumor survival, proliferation, and migration [7,8,9]. The classically activated myeloid cell phenotype is characterized by expression of surface CD86, stimulation of anti-tumor immune responses, and production of pro-inflammatory cytokines [10,11]. In contrast, TIMs are characterized by expression of CD206 and arginase 1 (Arg1) as biomarkers for the specific immune microenvironment, and these cells do not secrete cytokines definite for development of the innate immune response [10,11,12]. Instead, TIMs support tumor invasion, proliferation, angiogenesis, matrix remodeling, and resistance to ionizing radiation [7,13,14,15]. In this glioma–TIM crosstalk, glioma cells produce cytokines, such as CCL2, GM-CSF, and IL-6, that promote chemoattraction, recruitment, and maturation of myeloid cells in the tumor microenvironment [9,14]. These TIMs then release a wide array of factors, such as matrix metalloproteinases, IL-10, epidermal growth factor (EGF), vascular endothelial growth factor (VEGF), and IL-17, which are involved in the stimulation of proliferation and migration of tumor cells [8,13,16,17]. Normally, microglial activation of the central nervous system (CNS) confers protection from pathogens, removes debris, or performs structural remodeling; however, due to glioma-released cytokine influence, TIMs sustain an activated and polarized state that contributes to tumor proliferation and dispersal [7,13].

The TIM activation state is likely to have distinct impacts on tumor recurrence through the release of a specific pattern of chemokines. However, the immunological state of TIM in the brain tumor resection area is unclear. It was found that tumor-associated microglia/macrophages upregulate the release of transforming growth factor β (TGF-β), EGF, IL-10, and monocyte chemoattractant protein (MCP1, aka CCL2), leading to increased invasion and proliferation of glioma cells and provide a favorable environment for tumor growth [6,9,13,16,18]. Additionally, it was shown that the polarization state of microglia varies over time after brain injury [19,20]. The “on” signal that triggers microglial/macrophages activation in the wound area is conveyed by a range of molecules associated with cell damage [11,19]. It has been shown that in brain injury, microglia/macrophages at the injury site primarily undergo a polarization state to promote regenerative growth [20,21]. The tumor resection procedure creates a tumor and injury environment at one site. We hypothesize that tumor resection provides strong signaling for the polarization of myeloid cells in the resection site and exacerbates the malignant properties of tumor cells that fail to be eliminated by resection, providing a favorable environment for tumor recurrence. Targeting the TIM–glioma interaction during the post-surgical healing period may reduce the pro-tumorigenic effect of microglia/macrophages and delay tumor recurrence. Therefore, a clear understanding of the myeloid cell activation state at the tumor resection site and the role of the post-surgical microenvironment in mediating tumor relapse is vital. This study seeks to characterize the dynamics of the immunological state of cells of myeloid lineage in the tumor-resection area.

In this study, using the C57Bl6/GL261 mouse glioma implantation model, we found that TIM in tumors regrown after surgical resection undergo a dynamical shift associated with a significant increase in the Arg1-positive phenotype compared with primary implanted tumors. Through a combination of cytokine array and PCR analysis, we demonstrated upregulation in the expression of IL-10, IL-12, IL17, CCL2, and VEGF in TIM- infiltrating recurrent tumors compared with primary implanted tumors prior to the resection procedure.

## 2. Materials and Methods

### 2.1. Cell Culture

The GL261 glioma cell line was obtained from the National Cancer Institute (Frederick, MD, USA). Cells were cultured in Dulbecco’s modified Eagle’s medium (DMEM) supplemented with 10% fetal calf serum (Sigma-Aldrich, St. Louis, MO, USA), 0.2 mM glutamine, 50 U/mL penicillin, and 50 μG/mL streptomycin (Sigma-Aldrich, St. Louis, MO, USA) and maintained in a humidified atmosphere of CO_2_/air (5%/95%) at 37 °C. The medium was replaced with fresh culture medium about every 3–4 days.

### 2.2. Intracranial Glioma Tumor Implantation and Tumor Resection

C57Bl/6 mice were obtained from the Jackson Laboratory. GL261 glioma cells were implanted into the right cerebral hemisphere of 12–20-week-old C57BL/6 mice. Implantation was performed according to the protocol that we described earlier [22]. Briefly, mice were anesthetized with isoflurane, and a midline incision was made on the scalp. At stereotaxic coordinates of 2 mm lateral and 1 mm caudal to bregma, a small burr hole (0.5 mm diameter) was drilled in the skull. An aliquot (1 μL) of cell suspension (2 × 10^4^ cells/μL in PBS) was delivered at a depth of 3 mm over 2 min. The brain tumor was resected 14 days after implantation. The cavity was copiously irrigated, and the skin was closed. The animals were sacrificed at 0, 1, 4, 7, 14, and 21 days after resection, and the brain tissue was used for further studies. The IBox Explorer 2610 In Vivo Fluorescent Imaging System in combination with intracranial implantations of green fluorescent protein (GFP)–GL261cells and hematoxylin and eosin (H and E) staining of brain sections encompassing the tumor resection area were used to monitor the efficacy of resection.

### 2.3. Hematoxylin and Eosin Staining

Animals were anesthetized with pentobarbital (50 mg/kg, Sigma-Aldrich, St. Louis, MO, USA) and transcranially perfused with phosphate-buffered saline (PBS), followed by 4% paraformaldehyde (PFA). Animal brains were removed and postfixed in 4% PFA/PBS for 24 h at 4 °C, followed by 0.15 M, 0.5 M, and 0.8 M sucrose at 4 °C until fully dehydrated. Brains were then frozen-embedded in Cryo-M-Bed embedding compound (Bright Instrument, Huntingdon, UK), and 10-µm slides encompassing the tumor area were developed using a Vibratome UltraPro 5000 cryostat (American Instrument, Haverhill, MA, USA), followed by staining with H and E.

### 2.4. Western Blot Analysis

Clarified cell lysates separated by 10% SDS-PAGE were transferred to PVDF membranes and probed with an anti-arginase 1 (Arg1) antibody, developed in mice (cat. #SC-21738, Santa Cruz Biotechnology, Dallas, TX, USA), anti-CD86 (cat. #ab53004, Abcam, Boston, MA, USA), and anti-Iba1 (cat. #016-20001, Wako Chemicals USA Inc., Richmond, VA, USA), developed in rabbits, diluted 1:1000, and followed by addition of the corresponding secondary antibodies (cat. #A9169, Sigma-Aldrich, Saint Louis, MO, USA). Detection was performed with enhanced chemiluminescence methodology (cat. #34075, SuperSignal^®^ West Dura Extended Duration Substrate; Pierce, Rockford, IL, USA), and the intensity of the signal was measured using a gel documentation system (Versa Doc Model 1000, Bio-Rad, Hercules, CA, USA). β-tubulin (cat. #86298S, Cell Signaling, Danvers, MA, USA) immunoreactive signal was used as the loading control.

### 2.5. Myeloid Cell Purification from Tumor Tissue

To separate myeloid cells from a whole glioma tumor and healthy brain area, tissues were homogenized using a non-enzymatic cell-dissociation solution (Sigma-Aldrich, St. Louis, MO, USA) at 37 °C for 40 min. Cells were then washed with 1% PBS and passed through a 70-µm filter. Filtered cells were centrifuged at 100× *g* for 3 min, and microglial cells were purified using Percoll PLUS (cat. #1002914772, Sigma-Aldrich, St. Louis, MO, USA) gradients of 30%, 37%, and 70%. Following a 40-min centrifugation at 100× *g*, the distinct white ring of myeloid cells was collected at the 30%/37% Percoll interface.

### 2.6. Cytokine Antibody Array

To evaluate the expression of cytokines expressed in TIM, a Quantitative Mouse Cytokine Antibody Array (cat. #ab197465, Abcam, Boston, MA, USA) was used. Purified myeloid cells were lysed in cell buffer for 30 min, and the protein concentration was measured. Protein lysates were used for array hybridization according to the manufacturer’s protocol, and measurements of the fluorescent signals were performed in the Ray Biotech (Norcross, GA, USA) laboratory.

### 2.7. qRT-PCR

Gene expression of VEGF, CCL2, interleukins (IL-12, IL-17), and CD11b was analyzed using qRT-PCR. RNA was extracted from purified myeloid cells using the RNeasy Plus Mini Kit (cat # 74134, Qiagen GmbH, Hilden, Germany) following the manufacturer’s protocol. RNA quality and concentration were measured with the NanoDrop 1000 spectrophotometer (Thermo Scientific, Waltham, MA, USA). Complementary DNA (cDNA) was reverse-transcribed from total RNA using the iScript cDNA synthesis kit (Bio-Rad, Hercules, CA, USA). Asymmetrical cyanine dye SYBR green qRT-PCR gene expression assays were performed using VEGF, CD11b, IL-12, IL-17, and CCL2 primers (cat. #CED0040260, #CID0005641, #CIP0034996, and #CID0026592, Bio-Rad, Hercules, CA, USA). Amplification was carried out in a Bio-Rad CFX96 Touch real-time PCR detection system (Bio-Rad, Hercules, CA, USA). The gene expression level was defined as the threshold cycle number (CT). Mean fold changes in expression of the target genes were calculated using the comparative CT method (relative expression units, 2^−∆∆Ct^). All data were controlled for the quantity of input RNA by using GAPDH (cat. #1665010, Bio-Rad, Hercules, CA, USA) as the endogenous control and for normalization.

### 2.8. Flow Cytometry Analysis

TIM cells purified from brain tumors were washed with ice-cold PBS, counted by trypan blue exclusion, aliquoted to a concentration of 1 × 10^6^ cells per 100 μL, and blocked using 5 µg/mL anti-mouse CD16/32 (cat. #101320; Biolegend, San Diego, CA, USA) for 15 min at room temperature. Live cells were then immunolabeled with fluorophore-conjugated antibodies for 30 min on ice. Mouse anti-CD45–FITC, mouse anti-CD11b–PE, mouse anti-CD11c–APC, mouse anti-CD86–APC, and mouse anti-CD206–PE antibodies (cat. #1995400, #2010662, #1978183, #1995448, and #2005201, Thermo Fisher Scientific, Waltham, MA, USA) were used. Samples were then washed in Flow Cytometry Staining (FACS) buffer (1% BSA in PBS) and analyzed with a Guava^®^ easyCyte^™^ flow cytometer (Luminex Corporation, Austin, TX, USA) and Guava Incyte software (guavaSoft 3.3, Luminex Corporation, Austin, TX, USA). Cells were first gated for live cells using forward-scattered area (FSC-A) versus forward-scattered height (FSC-H) gating, followed by FSC/CD45 gating to identify CD45^low^ and CD45^high^ populations. Other expression markers were analyzed for each microglia/macrophage subset by gating on CD11b/CD11c or CD86/CD206 plots. Compensation was performed prior to each experiment using single-fluorophore-labeled compensation beads. Sample data acquisition was always preceded by running an unstained cell sample followed by a single-stained positive control to allow appropriate voltage adjustment.

### 2.9. Patient Datasets

Overall patient survival statistical data using Kaplan–Meier analysis of 540 glioblastoma cases from a Glioblastoma tumor RNA-Sequence dataset (TCGA-540-MAS5.0-u133a) were obtained from the R2 Genomic Analysis Visualization Platform. Available online: https://hgserver1.amc.nl/cgi-bin/r2/main.cgi (accessed on 1 July 2022).

### 2.10. Statistical Analysis

Results are expressed as mean ± standard deviation (SD). Statistical probability was calculated using 9GraphPad Prism 1.0 statistical software (San Diego, CA, USA). One-way analysis of variance (ANOVA) tests followed by Tukey’s post-hoc test was used to determine significance between groups. *p*-values of less than 0.05 were considered significant.

## 3. Results

### 3.1. Glioma Tumor Resection Modulates TIM Polarization

To determine the state of microglial polarization at the tumor resection site during healing after brain tumor resection, the C57BL/6/Gl261 mouse glioma implantation model was used. Two weeks after glioma implantation in mouse brain, the grown tumors were resected. At 0 h and 1, 4, 7,14, and 21 days after the tumor was resected, the animals were sacrificed, and the brain tissues from the area of resection were analyzed by Western blot to identify the expression levels of myeloid cell markers Iba1, CD86 and Arg1. cells. The brain cortex taken from the healthy contralateral hemisphere was used as a control to monitor total brain inflammation. Fluorescence-guided tumor resection with the use of GFP–GL261-implanted tumors and the IBox Explorer 2610 In Vivo Fluorescent Imaging System were employed to control, and assure the efficacy of, tumor resection (Appendix A). The dynamics of tumor regrowth after surgical resection was monitored with H and E staining of brain slices encompassing the tumor resection area. As presented in Appendix A, the signs of tumor regrowth were detected by day 7 after resection, and the tumors then gradually enlarged until day 21.

Figure 1 shows a Western blot analysis of CD86, Arg1, and Iba1 expression in the tumor-resected area and in the control healthy cortex area from the contralateral hemisphere for up to 21 days after tumor resection (Whole-membrane Western blot images and loading controls are presented in Appendix A). During the first 7 days after resection, a high expression level of CD86 together with a low expression level of Arg1 were detected at the tumor resection site, indicating an inflammatory activation of the tissue infiltrating myeloid cells. A switch in myeloid cell activation was observed by day 14, when a significant increase of Arg1 expression was identified in the tumor resection area and was further sustained up to the 21st day. By contrast, high expression of CD86, observed in the first seven days after resection, was significantly decreased by the 14th day. These results indicate the switch of tissue infiltrating myeloid cell activation at the resection site from a pro-inflammatory to an alternatively activated state in the period of 7–14 days after tumor resection. These dynamics of tissue-infiltrating myeloid cell activation were associated with tumor regrowth at the tumor resection site, which is demonstrated in Appendix A. Such inflammatory activation of myeloid cells immediately after tumor resection is likely associated with inflammation caused by tissue damage resulting from surgical intervention. The consequent healing of the wound, accompanied by tumor regrowth, redirected tissue-infiltrating myeloid cells into the polarization state, which is specific for TIM. Additionally, an increase in the Iba1 marker by day 14 (Figure 1A,B) indicates increased motility and the myeloid cell’s infiltration of tissue in relation to tumor regrowth. We did not observe significant changes in any of the three myeloid cell markers in the control healthy brain cortex, except for a slight upregulation of CD86 marker during the first days after surgery, indicating some post-surgical inflammation that was not detected by day 4. (Figure 1C,D).

Comparison of myeloid markers in primary implanted tumors (before resection) and regrown tumors (2 weeks after surgical resection) indicate that both tumors exhibit high Arg1 and low CD86 marker expression (Figure 1E,F). A significant (25%) upregulation of Arg1 in regrown tumors compared with primary implanted tumors was found. Of note, significantly lower expression of Arg1 and higher expression of CD86 were found in healthy cortex compared with tumor tissue. Taking into account the finding that Iba1 expression in primary implanted and regrown tumors did not significantly differ (indicating a similar level of myeloid cell infiltration), the elevated Arg1 in regrown tumors indicate a higher level of anti-inflammatory polarization, wound-healing, and pro-tumorigenic properties of TIM. These results provide evidence that primary GBM tumors and tumors regrown after resection harbor more pro-tumorigenic than pro-inflammatory TIM, which is associated with an impaired prognosis for GBM patients [23].

### 3.2. Cells of Myeloid Lineage Demonstrate Dynamic Variation in Cytokine Expression Profile at the Site of Tumor Resection

To further investigate modulations in the myeloid cell’s cytokine profile in the post-surgical period, myeloid cells purified from the tumor resection site were evaluated at time points of 0 h and 1, 4, 7, 14, and 21 days after tumor resection with the use of Quantitative Mouse Cytokine Antibody Arrays. This study found that IL-4, IL-5, IL-10, IL-17, CCL2, interferon gamma (IFN-γ), vascular endothelial growth factor (VEGF), and granulocyte–macrophage colony-stimulating factor (GM-CSF) were gradually upregulated through the post-surgical period and reached maximum expression on days 14 and 21 after tumor resection (Figure 2A). Considering that the 14- and 21-day time points correspond to the formation of regrown tumors, the results indicate that expression of the listed cytokines in regrown tumors is significantly higher than in primary implanted tumors. IL-4, IL-5, IL-10, IL-17, and CCL2 had elevated expression throughout the first day after resection and was sustained through formation of the regrown tumor by day 21. This finding indicates that expression of these factors was triggered by the resection procedure, supported during the healing period, and sustained by the newly regrown tumor. By contrast, IL-13 and IL-1a exhibited upregulation in the first days after resection, but this was later reversed back to the level of the primary implanted tumor, indicating the involvement of IL-1a and IL-13 in the inflammatory response caused by surgical intervention, but not in further tumor regrowth. VEGF and IL-12 exhibited upregulation from the 7th till 21st day after resection, which correlates with tumor regrowth after surgical resection. IL-6 displayed downregulation immediately after resection, with further restoration to the level of the primary implanted tumor. This suggests that IL-6 is basally expressed by TIM cells. Tumor resection eliminated TIM, but as tumors regrow, TIM express IL-6 again at the same level as in the primary implanted tumor.

These results demonstrate that surgical resection of tumors drastically modulates the myeloid cell’s cytokine expression profile at the tumor resection site. Three cytokine response subtypes were detected. First, cytokines that are elevated in response to the resection procedure and are probably related to tissue damage (IL-1a and IL-13). Expression of these cytokines reverts to the basal level with wound healing. Second, the group of cytokines were triggered in response to the resection procedure and then sustained in the newly regrown tumor (IL4, IL5, IL10, IL17, CCL2, IFN-γ and GM-CSF). Third, cytokines, whose expression is not driven by tissue damage but is later upregulated in regrown tumors, were probably triggered by other stimuli in the regrown tumor environment (VEGF and IL-12). In this way, the immunological state of TIM in regrown tumors significantly differs from that of the primary implanted tumor.

In the control cortex from the contralateral hemisphere, some signs of inflammation were detected on the first day after resection, including upregulation of IL-4, IL-6, IL-17, and CCL2 (Figure 2B). Whereas the levels of IL-4 and IL-6 reverted to basal levels by day 4, IL-17 and CCL2 sustained elevated levels until the 21st day after resection. Additionally, significant elevation of IL-1b expression in the healthy hemisphere was detected on day 21, corresponding to a fully regrown tumor. These results indicate that the tumor resection procedure and further tumor regrowth affect the immunological state of the whole brain in a persistent manner.

To further validate the dynamics of cytokines showing the greatest upregulation in regrown tumors, real-time qPCR was performed for myeloid cells purified from tumor-resected sites at indicated time points. Figure 3 shows a significant and gradual increase of IL-12 and IL-17 gene expression beginning the first day after tumor resection. Strong upregulation of VEGF and IL-12 gene expression was observed beginning on the 7th day after the resection procedure, which fully confirms the results obtained from the quantification of cytokine expression. Additionally, CD11b gene expression in purified myeloid cell populations showed an increase in the first days after resection and then returned to the pre-surgical level, indicating a short inflammatory response and relocation of myeloid cells to the wound area.

### 3.3. Primary Implanted and Regrown Tumors Harbor Various Myeloid Cell Populations

To evaluate whether the differences detected in cytokine expression profiles in myeloid cell fractions purified from primary implanted and regrown tumors are due to an immunological shift in TIM or due to differences in composition of myeloid cell populations, flow cytometry analysis was performed. CD45, CD11b, and CD11c markers were used to distinguish myeloid cell populations. Using FSC and CD45 plots, CD45^low^ (microglia) and CD45^high^ (peripheral myeloid cells) cell populations [24] were identified in primary implanted tumors, as well as in regrown tumors 1 and 2 weeks after resection (Figure 4A–C). No significant differences in the proportion of CD45^high^/CD45^low^ cells between primary implanted tumors and tumors 1- and 2-weeks post-resection were found (Figure 4D). 

Further analysis showed a slight shift in CD11b/CD11c marker expression in both CD45^low^ and CD45^high^ cell populations in weeks 1 and 2 post-resection compared with the primary implanted tumor. A gradual increase in CD45^low^/CD11b^high^/CD11c^low^ (microglia) and CD45^high^/CD11b^high^/CD11c^low^ (macrophages) [24] was detected in post-resected tumors compared with primary implanted tumors. Specifically, we detected 14.5 ± 4.1% CD45^low^/CD11b^high^/CD11c^low^ cells in primary implanted tumors, 15.5 ± 7.8% at 1 week post- resection, and 28.1 ± 7.5% in fully regrown tumors 2 weeks after resection (*p* < 0.05). Similarly, we detected 0.6 ± 0.4% CD45^high^/CD11b^high^/CD11c^low^ cells in primary implanted tumors, 1.1 ± 1.6% one week after resection, and 1.6% ± 0.6 in fully regrown tumors 2 weeks after resection (*p* < 0.05). These results indicate that tumors, regrown after surgical resection harbor a greater number of CD45^low^/CD11b^high^/CD11c^low^ and CD45^high^/CD11b^high^/CD11c^low^ cells than primary implanted tumors. Additionally, a large fraction of CD45^low^/CD11C^+^ cell population was detected (65–78% of total TAM in all investigated tumors) with a shift from the CD45^low^/CD11b^high^/CD11c^high^ to the CD45^low^/CD11b^low^/CD11c^high^ cell subpopulation in regrown tumors 2 weeks after resection relative to primary implanted tumors (29.5 ± 28.9% vs. 19 ± 18% and 48.6 ± 28% vs. 56.9 ± 38% respectively), and despite this shift, it did not reach statistical significance.

To additionally specify the TIM activation state, CD86 and CD206 markers were used for further analysis. A significant shift in CD86/CD206 expression was detected in TIM cells purified from primary implanted and regrown tumors. As presented in Figure 5, most cells in primary implanted tumors represent the CD206^high^/CD86^low^ phenotype for both CD45^low^ and CD45^high^ subtypes (81.9 ± 13.3% for CD45^high^ and 76.2 ± 7.9% for CD45^low^). One week after tumor resection, the CD86^high^/CD206^low^ phenotype was mostly detected at the resection site (87.1 ± 2.7% in CD45^high^ and 89.3 ± 13.3% in CD45^low^ subtypes). Two weeks after tumor resection, myeloid cells infiltrated regrown tumors, demonstrating another shift toward predominant expression of CD86^high^/CD206^high^ (84.1 ± 6.3% in CD45^high^ and 68.6 ± 6.2% in CD45^low^ subpopulations). 

Summarizing, despite the fact that the composition of myeloid cells varies only modestly in primary implanted and regrown tumors, the activation state of tumor-infiltrating myeloid cells undergoes significant modulation in regrown tumors.

### 3.4. Cytokines Expression Pattern and Myeloid Cell Markers in Glioblastoma Tumors Associatde with Overall Patient Survival

R2 Genomic Analysis Visualization Platform was used for the analysis of expression of identified cytokines in GBM tumors on overall patient survival. Kaplan–Meier analysis of 540 glioblastoma cases from the Glioblastoma tumor RNA-Sequence dataset we identified showed a negative association between levels of gene expression of IL-4, IL-10, CCL-2, GM-CSF, and VEGF and overall survival (Appendix A). The 24-month overall survival probability in a cohort of patients with high gene expression versus low gene expression of listed chemokines in tumor specimens was 0.2 vs. 0.35 for IL-4, 0.2 vs. 0.35 for IL-10, 0.2 vs. 0.4 for CCL2, 0.2 vs. 0.57 for GM-CSF, and 0.15 vs. 0.23 for VEGF. Association between high gene expression and 24-monthsoverall survival probability was identified for IL-5 (0.1 for low gene expression vs. 0.25 for high expression), IL-12 (0.2 vs. 0.4), and IFN-γ (0.18 vs. 0.28). No effect of IL-17 gene expression on overall survival probability was found; however, association with progression free survival probability was identified as a 0.35 progression-free survival probability 24 months after tumor resection in patients with low expression of IL-17 in tumors vs. 0.2 in tumors with high expression.

Analysis of myeloid cell markers CD86 and Arg1 revealed a distinct association with a patient’s overall survival probability: high CD86 gene expression and low expression of Arg1 were related to shorter overall survival, whereas low CD86 and high Arg1 were related to low overall survival (Appendix A). Furthermore, a 0.15 overall survival probability was identified for patients with high CD86 gene expression in tumor specimens vs. 0.3 for low expression, and 0.25 vs. 0.17 correspondingly in patients with high and low Arg1 gene expression in tumors.

## 4. Discussion

It is noteworthy that surgical resection remains one of the first lines of GBM treatment, yet it is the least-studied component of GBM therapy in term of its effect on the immunological state at the site of resection and on the tumor microenvironment. Furthermore, TIM activation at the resected site plays an essential role in the pathogenesis of glioma and elucidating the TIM polarization state may shed light on novel therapeutic approaches for brain tumor immunotherapy. In this study, we found that tumor resection has an immediate and prolonged effect on the activation state and cytokine expression profile of glioma-associated myeloid cells. The tumor resection procedure causes a dramatic and immediate inflammatory response at the tumor resection site, accompanied by upregulation of the CD86 marker. The further dynamics of CD86 and Arg1 expression at the tumor resection site, as revealed by Western blot analysis, correlates with tumor regrowth after surgical resection, indicating that tumor removal eliminates tumor-derived signaling, driving myeloid cells into the polarization state, which is characteristic for TIM. However, regrowing a tumor creates a microenvironment that directs myeloid cells toward expression of tumor specific polarization markers [14,23]. Additionally, the final level of Arg1 expression in fully regrown tumors was upregulated by 25% compared with primary implanted tumors, indicating a switch in arginine metabolism, an increase in matrix deposition capacity in TIM, and a correspondingly higher pro-tumorigenic activity (Figure 1E,F) [7,25]. Statistical analysis of overall survival probability (Appendix A) identified direct association between high gene expression of Arg1 and low overall patient’s survival. This indicates that elevated expression of Arg1 in regrown tumors may result in more aggressive growth of relapse tumors compared to newly diagnosed tumors and worser disease progression prognoses. In accordance with these results, flow cytometry analysis displayed a similar expression pattern in primary implanted and regrown tumors, indicating a shift toward CD86 marker expression shortly after tumor resection and then another shift towards high expression of both CD206 and CD86 in fully regrown tumors (Figure 5). Previous studies have reported that TIM express a mixture of genes and markers in advanced glioma tumors [26,27], which reflect two types of inflammation: T helper (Th1)-related type I inflammation leading to natural killer (NK) cell activation through production of tumor necrosis factor α (TNFα), IL-6, IL-8, and IL-12, and Th2-response type II inflammation, which induces regulatory T cells (Treg) through the production of IL-10 and VEGF, Th2 immunoregulation, matrix deposition, and tissue remodeling [13,27]. Although some discrepancy exists in the exact activation state of TIM, that is, whether there is a higher accumulation of pro-inflammatory or pro-tumorigenic TIM in the tumor area, it has been demonstrated that a heterogeneous population of TIM in both activated states can coexist in the core of the tumor [28,29]. Our results support the conclusion that tumor resection removes the GBM tumor core, which promotes an immunosuppressive environment, but later, as the glioma tumor regrows after surgical resection, it harbors TIM with the characteristics of a mixed pro- and anti-inflammatory polarization.

Cytokine array analysis showed that, during the post-surgical period at the tumor resection site, myeloid cell population exhibit a highly dynamic cytokine expression profile (Figure 2). Based on the dynamics of expression, all myeloid cell cytokines can be divided into three groups: (1) those upregulated at the site of tumor resection as an immediate response to surgical intervention and then downregulated with healing of the cavity (IL-1α and IL-13), (2) those upregulated after tumor resection and then having sustained upregulation during tumor regrowth (IL-4, IL-5, IL-10, IL-17, CCL2, GM-CSF, and IFNγ), and (3) those demonstrating delayed elevation of expression correlated with tumor regrowth (VEGF and IL-12). Modulation of cytokines from the first group is transient and likely related to tissue damage at the resection site. However, the expression of cytokines listed in group (2) are triggered by the surgical procedure, and the elevation is sustained in the regrown tumor, which results in significant modulation of the tumor microenvironment. Several studies have shown that IL-4 serves as an anti-inflammatory cytokine in the tumor microenvironment. It has been observed that IL-4 in combination with IL-13 induces macrophages to support tumor survival by creating an immunosuppressive environment and enhancing tissue repair [27,30]. Toll-like receptor (TLR)-induced cytokine production is impaired in IL-4-exposed microglia, leading the inhibition of microglial innate immune responses [31].

Additionally, IL10, which was found in our study to be upregulated in regrown tumors compared with primary implanted tumors, was shown to promote glioma cell growth and invasion [32,33] Moreover, IL-10 was shown to suppress proliferation, cytokine production, and migration of effector T-cells, driving T-cell exhaustion and thus creating an immunosuppressing environment in the tumor [34,35]. Increased expression of IL-10 in GBM patients was correlated with brief survival [36]. IL-4 and IL-10 are cytokines that induce pro-tumorigenic anti-inflammatory polarization of TIM and are related to low overall survival probability (Appendix A), and our study found upregulation of both in tumors regrown after surgical resection.

Upregulation of IL-17 was found in GBMs; however, reports about the association of prognostic patient survival with IL17 expression are contradictory. The Madkouri group found an association of IL-17 with poor survival [37], whereas others found that high levels of IL-17 expression in the tumor tissues were a good prognostic marker for patients with GBM [38]. The same variable association of IL-17 expression and patient survival was shown for non-brain tumors as well. Poor survival was associated with high IL-17 in cervical cancers [39] and hepatocellular carcinoma [40] but was found to be a favorable prognostic factor for esophageal squamous cell carcinoma [41]. IL-17 has been found to promote migration, extracellular matrix degradation, and invasion of cancer cells, including GBM cells, through the activation of PI3K–AKT [42,43] and STAT3 signaling [43,44], as well as through the elevation of expression of the matrix metalloproteinases MMP-2 and MMP-9 [42]. It was observed in hepatocellular carcinoma that IL-17-driven AKT signaling activation resulted in IL-6 production, which in turn activated JAK2–STAT3 signaling and subsequently upregulated its downstream targets, IL-8, MMP2, and VEGF [43]. In this way, IL-17 expression in tumors is associated not just with tumor invasiveness but also with increased VEGF expression, which was observed in our study in the later steps of tumor regrowth after resection, suggesting an increased vascularity in regrown GBM tumors. Additionally, VEGF was found to contribute to tumor growth [14], promote the mobilization of myeloid cells to the tumor area [45], modulate the innate immune response in GBM [46], and is associated with poor patient overall survival probability (Appendix A). Despite not identifying any impact of high or low expression of IL-17 in GBM tumors on overall survival probability in our study, the strong negative association between the level of IL-17 gene expression and progression free survival probability was found (Appendix A).

Our study has established the upregulation of CCL2 and GM-CSF in regrown tumors compared with primary implanted tumors, together with poor overall survival probability in patients with high CCL-2 and GM-CSF expression in tumors. CCL2 is known to promote cancer development by facilitating the attraction of microglia, circulating monocytes, and myeloid-derived suppressor cells to the tumor area [9,10,47], whereas GM-CSF promotes the proliferation of TIM [48]. Glioblastoma cells can themselves produce CCL2 and GM-CSF and promote the attraction of TIM [9,48]. Once inside the tumor, myeloid cells undergo polarization toward TIM phenotype and release additional MCP-1, which forms an amplification loop helping to maintain a significant number of TIM in the tumor environment and reprogramming the tumor microenvironment into an immunosuppressive state [49,50]. In addition, overexpression of CCR2A (the receptor for CCL2) was detected in human GBM specimens compared with normal brain tissues and was associated with increased migration of glioma cells [51]. In support of this finding, it was observed that CCL2 facilitates migration and metastasis in ovarian cancer by activating the PI3K–AKT–mTOR pathway and its downstream effectors, HIF-1α and VEGF-A [52,53]. Taken together, the upregulation of CCL2 and GM-CSF in regrown tumors detected in our study suggests increased TIM infiltration and proliferation in relapsed tumors, an immunosuppressive shift in the tumor microenvironment, and enhanced glioma cell motility.

IL-12 is a potent immunostimulatory cytokine that has been extensively investigated in cancer therapy and has been shown to increase survival in rodent glioma models [54,55]. Our study identified increased overall survival probability in patients with high expression of IL-12 and IFN-γ in tumors (Appendix A). IL-12 facilitates maturation of natural killer cells and cytotoxic T cells; stimulates secretion of IFN-γ, leading to anti-angiogenesis; and supports the adaptive immunity of tumors [56]. Our study revealed upregulation of IL-12 in the later steps of development of the recurrent tumor. Taking into account the polarization shift of TIM toward a mixed activation phenotype during regrown tumor formation, the involvement of IL-12 and the related expression of IFN-γ can be proposed as a TIM switch to a mixed activation polarization. To clarify the role of IL-12 in the advanced steps of recurrent GBM tumors, further studies are needed.

Previous studies indicated strong overexpression of IL-6, IL-10, VEGF, GM-CSF, and CCL2 in human GBM specimens, and the level of overexpression strongly correlated with tumor grade, proliferation markers, and clinical aggressiveness in glioblastomas [57]. Summarizing the modulations in the cytokine expression profile (Figure 6), we propose that there is a significant shift in TIM cytokine expression pattern in recurrent tumors compared with primary implanted tumors. The most significant upregulation of expression in regrown tumors was identified for VEGF, IL-12, IL-17, GM-CSF and CCL2, which was confirmed at both the transcriptional and translational levels (Figure 2 and Figure 3). Strong elevation of these cytokines indicates increased proliferation and invasion of glioma cells, increased vascularization of the tumor, and an immunosuppressive environment in combination with an inflammatory component [45,49,54]. Elevated treatment resistance seems likely, taking into account the elevated level of IL-17 and related Akt signaling activation, as reported previously [42,43]. Taken together, the tumor microenvironment in regrown tumors significantly differs from primary implanted tumors and supports more aggressive tumor behavior and resistance to therapy.

A slight modulation in cytokine expression was detected in myeloid cells purified from the healthy contralateral hemisphere. Upregulation of IL-4, IL-5, and IL-6 was detected in the healthy hemisphere in the first days after tumor resection, which later reverted to basal level, indicating a transient in-brain inflammatory response to the surgical intervention. The level of CCL2 and IL-17 increased from day 1 after tumor resection and stayed elevated through day 21, indicating sustained post-surgical activation of brain myeloid cells. Sustained elevation of MCP1 was detected in the cerebrospinal fluid of traumatic brain injury patients [58] and in acute excitotoxic injury in the neonatal mouse brain [59]. Additionally, CCL2 was shown to induce neural stem cell proliferation and differentiation [60]. Together with these effects, high levels of IL-17 were detected in a number of inflammatory conditions in the CNS, including cancer [61], which suggests that CCL2 and IL-17 are involved in regulation of the whole-brain response to post-surgical and tumor-development conditions and might be involved in compensatory neuronal network remodeling after brain damage caused by surgical intervention and tumor regrowth. The increase in IL-1a in the healthy hemisphere by the 21st day detected in our study correlates with tumor regrowth and might be related to the inflammatory response affecting the entire brain as the tumor grows.

Flow cytometry analysis did not show significant differences in CD45^high^ and CD45^low^ cell populations in primary implanted and regrown tumors (Figure 4D), indicating that the microglial and peripheral myeloid cell population ratios are the same in both primary implanted and regrown tumors. Microglia are classically defined as CD45^low^, and macrophages and dendritic cells (DC) are defined as CD45^high^-expressing cells [24]. However, it was found that myeloid cells, including microglia, increase the expression and activity of CD45 in hypoxic conditions [62] and can upregulate CD45 upon cultivation in tumor-conditioned medium [63]. Considering the fact that hypoxia is a hallmark of glioma and that previous studies found that in gliomas microglia are not only restricted to the CD45^low^ fraction but can also be found in the CD45^high^ population [64], some of the CD45^high^ cells found in our study could belong to the microglial population rather than to the population of infiltrating macrophages.

When analyzing CD11b and CD11c markers to further distinguish CD45^+^ cells, no distinct cell populations were identified, indicating that tumor-infiltrating myeloid cells express varying levels of CD11b and CD11c. A shift toward CD45^low^/CD11b^high^/CD11c^low^ (microglia) and CD45^high^/CD11b^high^/CD11c^low^ (macrophage) expression markers in the myeloid cell population, observed in regrown tumors, indicates increased accumulation and proliferation of microglia and macrophages, which correlates with the upregulation of CCL2 and GM-CSF in the regrown tumors, as detected in our study. Upregulation of CD45^low^/CD11b ^low^/CD11c^high^ cells, defined as CD11c^+^ microglia, might be a sign of modification in the immunological state of microglia in regrown tumors, resulting in changes in cytokine expression profile, as determined in this study. Upregulation of CD11c-positive microglia, which make up to 30% of total activated microglia, was found for a number of brain inflammatory conditions resulting from brain damage [65], autoimmune disorders [66], and Alzheimer’s disease [67]. The CD11c^+^ microglial subset was found predominantly in primary myelinating areas of the developing brain and expresses genes for neuronal and glial survival, migration, and differentiation [68]. A comparison between the transcriptomes of isolated CD11c^+^ and CD11c^−^ microglia showed that gene expression had a similar general pattern, except for genes involved in immune signaling and lysosome activation, suggesting a suppressive/tolerizing influence by CD11c^+^ cells [67]. In contrast to CD11c, the integrin CD11b upon activation promotes pro-inflammatory microglia and macrophage polarization, whereas inhibition of CD11b promotes an immunosuppressive state in microglia and is related to accelerated tumor growth [69,70]. Since TIM downregulate the expression of genes involved in the inflammatory state phenotype [71], the abundant population of CD45^low^/CD11b^low^/CD11c^high^ cell populations (48% and 56% in primary implanted and regrown tumors, respectively) reflects the immunosuppressive environment in glioma. In general, a shift toward a CD11b^high^/CD11c^low^ phenotype in both CD45^high^ and CD45^low^ populations, together with upregulation of CD11c-positive microglia identified in regrown post-resection tumors (Figure 4), correlates with upregulation of the CD86^high^/CD206^high^ population (Figure 5) and an altered cytokine expression profile (Figure 2), indicating acquisition of a mixed pro- and anti-inflammatory immunological state in regrown tumors. However, as no proportional difference between microglia and peripheral macrophages was found infiltrating primary implanted and regrown tumors, this shift is likely to be the consequence of an altered microenvironment in the regrown tumor.

Our data are consistent with previous studies, performed with use of orthotopic xenograft tumor resection model in rats and demonstrated that recurrent tumors were more proliferative compared with primary tumors, and upregulated VEGF and interleukin signaling harbored a higher proportion of myeloid cells [72]. However, although our study did not find differences in CD45^high^ and CD45^low^ cell populations in primary implanted and regrown tumors, the Movahedi group demonstrated elevation of monocyte-derived macrophages in human recurrent GBM tumors compared with newly diagnosed tumors [73]. This discrepancy might be explained with exposure of recurrent tumors with chemo and radiotherapy, leading to loss of microglia with further replacement with macrophages.

Our previous studies and literature reports demonstrated that GBMs are immunologically heterogeneous [17,74]. It was shown in murine models that myeloid cell composition in tumors generated with different driver mutations showed marked differences: Nf1-silensed tumors had the highest level of Iba1 immunopositivity, compared to the PDGFB-overexpressing and EGFRvIII tumors [74]. This observation was confirmed for their corresponding human GBM subtypes. Additionally, the same study indicated that the majority of myeloid cells in PDGFB-overexpressing and EGFRvIII-expressing tumors were monocyte-derived, whereas in Nf1-silenced murine tumors, the majority of the myeloid cells were brain-resident microglia. A similar trend was described in human GBM samples [23]. Based on these data, using one animal model represents a limitation of this study. The TIM composition and dynamic of cytokine expression profile in tumor resection site in post-surgical period in tumors with distinct genetic profile should be further investigated.

## 5. Conclusions

In conclusion, we found that in the GL261/C57Bl/6 glioma implantation model, the tumor resection procedure results in modulation of the activation state and cytokine expression profile of TIM in recurrent tumors compared with primary implanted tumors. The surgical procedure activates an inflammatory response by TIM at the resection site, leading to sustained upregulation in expression of IL-4, IL-5, IL-10, IL-12, IL-17, VEGF, CCL2, and GM-CSF in regrown tumors. CD86^+^/CD206^+^ microglia/macrophages were identified in regrown tumors, contrary to the presence of CD86^−^/CD206^+^ in primary implanted tumors. Identified sustained upregulation of the Arg1 marker in myeloid cells, purified from regrown tumors in the Gl261/C57Bl/6 mouse animal model, was found to be associated with low overall survival probability in GBM patients. No significant difference in the ratio of tumor infiltration with microglia vs. peripheral macrophages was found, indicating that the detected difference in cytokine expression profile is due to modulations of polarization of TIM, but not due to a difference in myeloid cell populations infiltrating primary implanted and regrown tumors.

## Figures and Tables

**Figure 1 brainsci-12-00893-f001:**
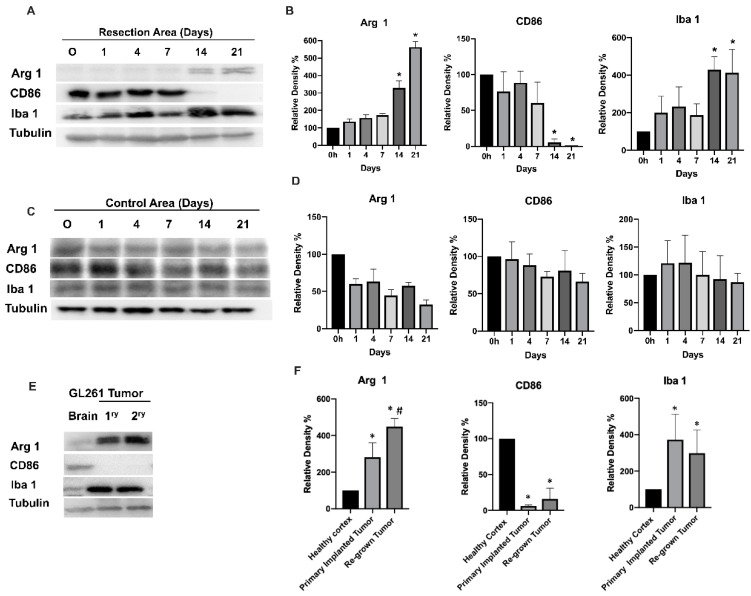
Western blot analysis (**A**,**C**,**E**) and quantification (**B**,**D**,**F**) of myeloid cell markers at the brain tumor resection site and in control cortex from the contralateral hemisphere. The Gl261/C57Bl/6 mouse glioma implantation model was used. Time points 0, 1, 4, 7, 14, and 21 days after tumor resection are shown for the tumor resection site (**A**,**B**) and for healthy cortex from the contralateral hemisphere (**C**,**D**). Comparative expression of myeloid cell markers is shown for healthy cortex in the contralateral hemisphere for a primary implanted tumor (1ry) and for a tumor regrown after resection (2ry; **E**,**F**). Mean + S.E., and significant differences between groups are shown (*p* < 0.005). *, a significant difference from 0 h after resection (**B**,**D**) or to healthy cortex (**F**); #, a significant difference between primary implanted (1ry) and regrown (2ry) tumors. (*N* = 5).

**Figure 2 brainsci-12-00893-f002:**
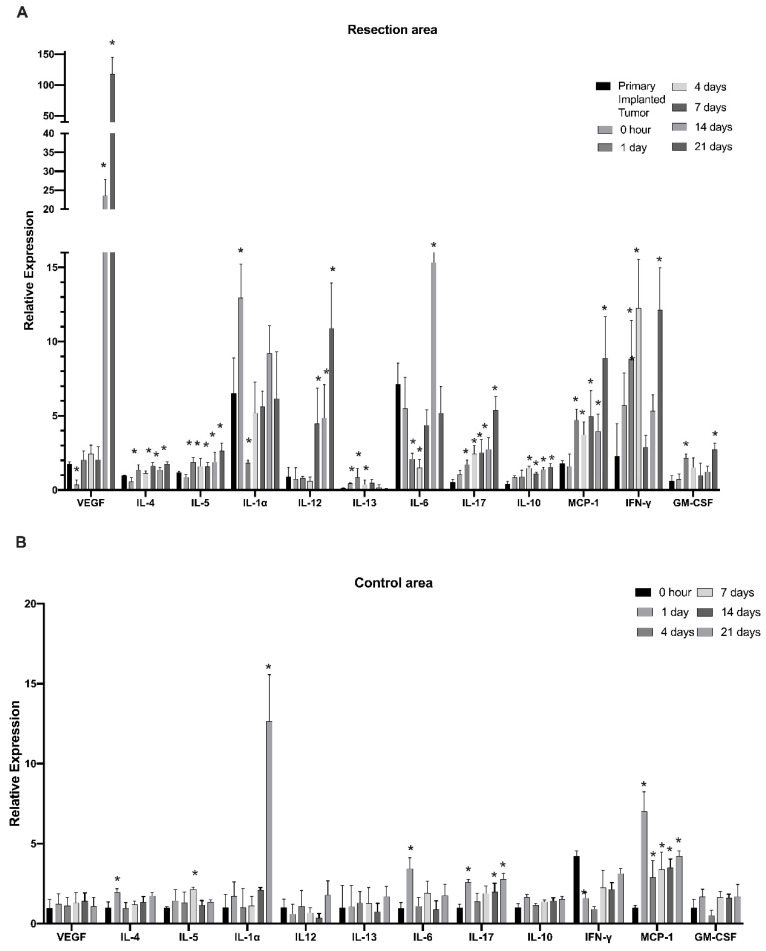
Dynamics of cytokine expression in myeloid cells at the infiltration site of tumor resection in the GL261-C57BL/6 glioma implantation mouse model. The relative expression of cytokines in the tumor resection area (**A**) and cortex tissue taken from the control contralateral hemisphere (**B**) at 0, 1, 4, 7, 14, and 21 days after resection. Mean + S.E., and significant differences from the primary implanted tumor (**A**) or cortex tissue taken before resection (**B**) are indicated (* *p* < 0.005), *N* = 3.

**Figure 3 brainsci-12-00893-f003:**
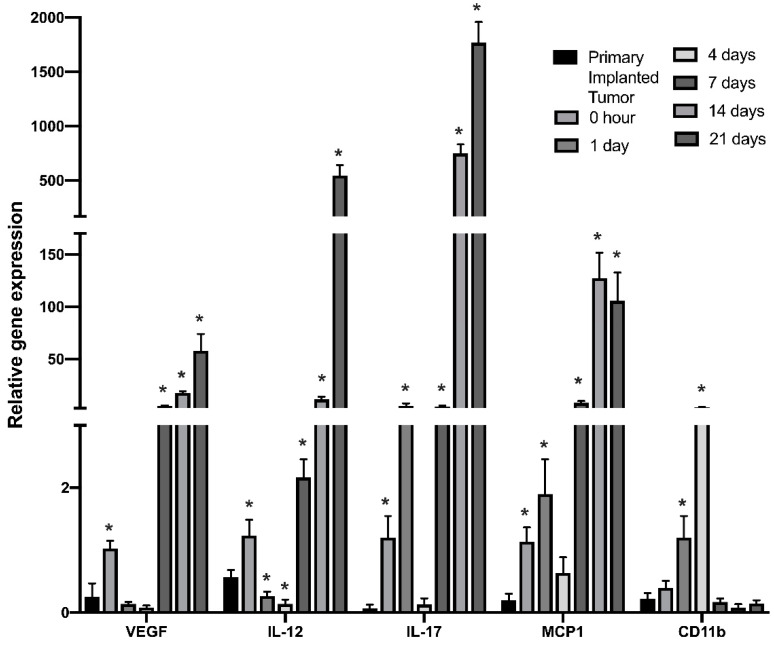
Real-time PCR analysis of cytokine gene expression in cells of myeloid lineage isolated from the tumor-resected area in the GL261/C57Bl/6 mouse glioma implantation model. RT-PCR analysis was performed for myeloid cells purified from tumors before resection and from the tumor resection site at 0, 1, 4, 7, 14, and 21 days after surgical resection. Mean + S.E., and significant differences from primary implanted tumor before resection (* *p* < 0.005) are shown (*N* = 3).

**Figure 4 brainsci-12-00893-f004:**
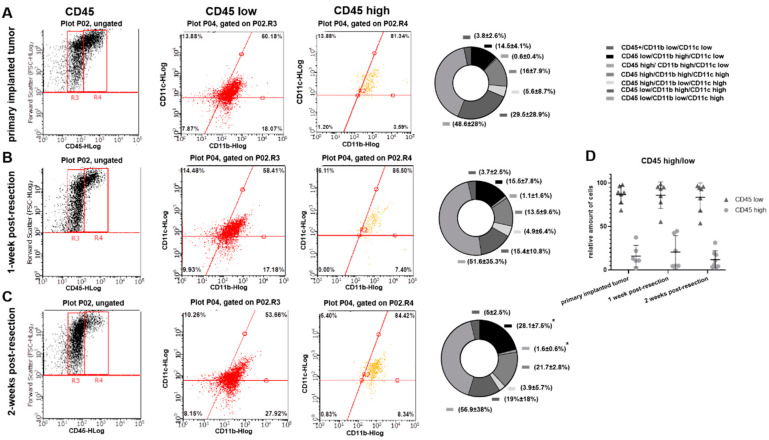
Flow-cytometric analysis of myeloid cells in primary implanted tumors and tumors 1- and 2-weeks post-resection in the GL261-C57BL/6 mouse glioma implantation model. (**A**–**C**) The left column contains forward scatter (FSC) vs. CD45 plots demonstrating CD45 populations: low (R3) and high (R4). The middle column contains CD11b^+^ vs. C11c^+^ events plotted from CD45^high^ and CD45^low^ populations. Graphs showing proportional composition of cellular subtypes are shown in the right column. (**D**) Calculation of CD45^high^ and CD45^low^ populations of microglia/macrophages purified from primary implanted tumors and tumors 1- and 2-weeks post-resection. Mean + S.D., and significant differences from the primary implanted tumor (* *p* < 0.005) are shown (*N* = 6).

**Figure 5 brainsci-12-00893-f005:**
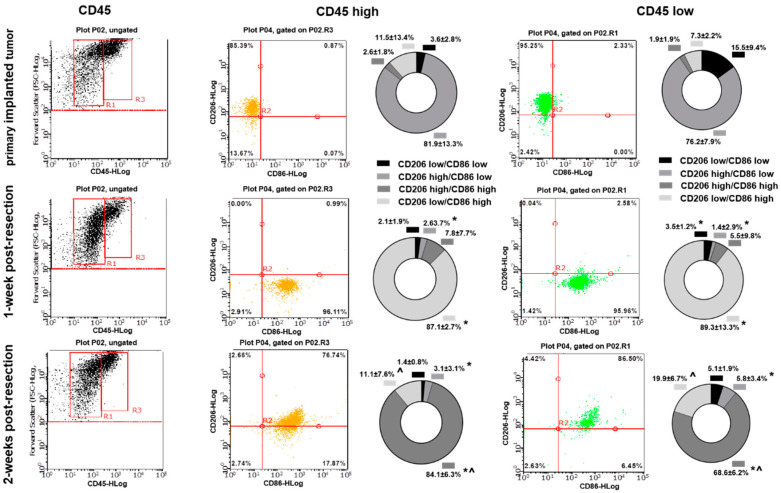
Flow-cytometric analysis of CD86 and CD206 expression in tumor-infiltrating myeloid cells purified from primary implanted tumors and tumors regrown 1 and 2 weeks after resection in the GL261-C57BL/6 mouse glioma implantation model. The left column contains forward scatter (FSC) vs. CD45 plots demonstrating CD45 populations: low (R1) and high (R3). The middle column contains CD86 vs. CD206 events plotted from CD45^high^ and CD45^low^ populations. Graphs showing proportional composition of myeloid cell markers are shown in the right columns. Mean + S.E., and significant differences from primary implanted tumors (*) and from 1-week post-resected tumors (^) are shown (*p* < 0.005, *N* = 6).

**Figure 6 brainsci-12-00893-f006:**
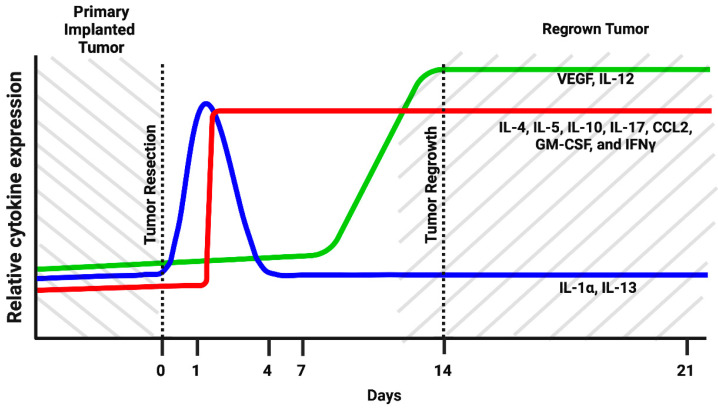
Schematic representation of the dynamic changes in myeloid cells cytokine expression pattern in tumor resection site during postsurgical period in the GL261-C57BL/6 mouse glioma implantation model. (Created with BioRender.com (accessed on 28 June 2022)).

## Data Availability

The data generated in this study are available within the article and its Appendix A.

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
