# Peer review of "The Dynamics of Tumor-Infiltrating Myeloid Cell Activation and the Cytokine Expression Profile in a Glioma Resection Site during the Post-Surgical Period in Mice"

_brainsci, 2022, doi:10.3390/brainsci12070893_

Round 1
Reviewer 1 Report
This is a well-written manuscript that investigated the profile of the tumor infiltrated immune cells prior to and post tumor resection in the immunocompetent mouse glioblastoma model. A wide variety of myeloid cell biomarkers is analyzed, revealing a significant increase in expression of CD86 during the first seven days after surgical tumor resection and then upregulation of the expression of arginase 1. The primary and recurrent tumors had different patterns of tumor infiltrated cells and cytokine production. Overall, the manuscript is interesting to readers and improves understanding of the role of tumor microenvironment in glioblastoma recurrence. Authors found that the tumor resection provides a strong pro-tumorigenic initially pro-inflammatory effect and the induction of strong signaling for the polarization of myeloid cells in the resection site. I recommend the manuscript for publication with minor revision. I have a couple of suggestions:
1. My recommendation will be to analyze the impact of expression of analyzed cytokines and myeloid cell biomarkers on overall patient survival (by utilizing Kaplan Meier plots) in partially and completely resected glioblastomas using R2: (R2: Genomics Analysis and Visualization Platform database, French gliomas) and include this analysis in the supplemental materials.
2. Please include in the discussion that the neutrophil subset of myeloid-derived cells (Ly6ghighCD45high of CD11b+CD45high) was not analyzed in the current work; however, it may play a significant role in the pro-inflammatory state after tumor resection and tumor recurrence (1, 2).
1. Rahbar A., Cederarv M., Wolmer-Solberg N., Tammik C., Stragliotto G., Peredo I., Fornara O., Xu X., Dzabic M., Taher C., et al. Enhanced neutrophil activity is associated with shorter time to tumor progression in glioblastoma patients. Oncoimmunology. 2015;5:e1075693.
2. Emilie Le Rhun, Felix Boakye Oppong, Maureen Vanlancker, Roger Stupp, Burt Nabors, Olivier Chinot, Wolfgang Wick, Matthias Preusser, Thierry Gorlia, Michael Weller, Prognostic significance of therapy-induced myelosuppression in newly diagnosed glioblastoma, Neuro-Oncology,2022; noac070, https://doi.org/10.1093/neuonc/noac070t
Author Response
We thank the reviewer for the suggestion to use R2 Genomics Analysis and Visualization Platform. This is a user-friendly platform encompassing a large number of datasets and provides broad opportunities for extended analysis.
- My recommendation will be to analyze the impact of expression of analyzed cytokines and myeloid cell biomarkers on overall patient survival (by utilizing Kaplan Meier plots) in partially and completely resected glioblastomas using R2: (R2: Genomics Analysis and Visualization Platform database, French gliomas) and include this analysis in the supplemental materials.
We added patient dataset analysis to the results section (lines 404-424) as well as to the Discussion (lines 443-447, 485, 506-509, 512, 529). Kaplan-Meier plots are now provided in Supplementary Fig.3 and 4. In addition to this, we provided a reference from the multiplex study, identified significant up-regulation of IL-6, IL-1β, TNF-α, IL-10, VEGF, FGF-2, IL-8, IL-2, and GM-CSF in human GBM specimens, which correlated with tumor grade, proliferation markers, and clinical aggressiveness in glioblastomas (lines 539-541).
- Please include in the discussion that the neutrophil subset of myeloid-derived cells (Ly6ghighCD45high of CD11b+CD45high) was not analyzed in the current work; however, it may play a significant role in the pro-inflammatory state after tumor resection and tumor recurrence (1, 2).
- Rahbar A., Cederarv M., Wolmer-Solberg N., Tammik C., Stragliotto G., Peredo I., Fornara O., Xu X., Dzabic M., Taher C., et al. Enhanced neutrophil activity is associated with shorter time to tumor progression in glioblastoma patients. Oncoimmunology. 2015;5:e1075693.
- Emilie Le Rhun, Felix Boakye Oppong, Maureen Vanlancker, Roger Stupp, Burt Nabors, Olivier Chinot, Wolfgang Wick, Matthias Preusser, Thierry Gorlia, Michael Weller, Prognostic significance of therapy-induced myelosuppression in newly diagnosed glioblastoma, Neuro-Oncology,2022; noac070, https://doi.org/10.1093/neuonc/noac070t
Neutrophil activation in blood samples is an important early sign of tumor progression in GBM patients. Lower neutrophil counts were associated with better overall patient’s survival, as it was demonstrated in publications listed above. However, despite clear detection of Ly6G+/Lys6Cinter cell population (PMN-MDSCs and/or neutrophils) within bone marrow isolates in mouse glioma implantation models, Ly6G+/Lys6Cinter cells are rarely evident in other murine glioma models, i.e. KR158 or 005 GSC tumors (Flores-Toro et al, 2020). Based on this experience with mouse glioma implantation models, it is unlikely that GL261 tumors contain a significant population of these cells. In this study we analyzed myeloid cells in brain tumor tissue, but not in blood or bone marrow. Though, the role of circulating neutrophils in GBM tumor recurrence is of definite medical and scientific interest and should be investigated in greater detail, we did not incorporate discussion of the neutrophil population in this report to keep it focused on monocytic myeloid populations (microglia, macrophages, MDSCs).
Flores-Toro JA, Luo D, Gopinath A, Sarkisian MR, Campbell JJ, Charo IF, Singh R, Schall TJ, Datta M, Jain RK, Mitchell DA, Harrison JK. CCR2 inhibition reduces tumor myeloid cells and unmasks a checkpoint inhibitor effect to slow progression of resistant murine gliomas. Proc Natl Acad Sci U S A. 2020 Jan 14;117(2):1129-1138. doi: 10.1073/pnas.1910856117.
Reviewer 2 Report
Ortiz- Rivera et al present an interesting study using an immunocompetent mouse glioma model to study the dynamic changes in cytokine expression after surgical resection. This knowledge may be useful in deciding the optimal timing to use immune checkpoint inhibitors in malignant gliomas. Although the study is interesting, the following points need to be addressed before consideration for publication.
Major points:
1. Conclusions cannot be proposed from data of only one model (cell line). At least four other immunocompetent mouse models (GL26 (C57BL/6) CT-2A (C57BL/6), SMA-560 (VM/Dk), and 4C8 (B6D2F1)) have been reported and immunocompetent glioma models can also be achieved through short-term, systemic costimulation blockade strategy (CTLA-4-Ig and MR1). Please provide at least two models.
2. I am curious to see how close these mouse models are to actual human surgical tissues. Can the authors show cytokine data for Day 0 after surgical resection in surgically obtained human tissue, or at the least discuss previous studies looking at this issue?
3. It would be great if the authors could provide a figure illustrating the dynamic changes in cytokine expression at the different time points after surgery.
Minor points:
1. Nation Cancer Institute (line 99) should be National Cancer Institute.
Author Response
We thank the reviewer for the suggestion to analyze alternative glioma models and immunological heterogeneity of GBMs in relation to tumor genetic profile. It helped us to strengthen the manuscript. New figure depicting the dynamic changes of cytokine’s expression in postsurgical period is now added as Fig. 6 and helps visually summarize the study results.
Major points:
- Conclusions cannot be proposed from data of only one model (cell line). At least four other immunocompetent mouse models (GL26 (C57BL/6) CT-2A (C57BL/6), SMA-560 (VM/Dk), and 4C8 (B6D2F1)) have been reported and immunocompetent glioma models can also be achieved through short-term, systemic costimulation blockade strategy (CTLA-4-Ig and MR1). Please provide at least two models.
GBMs are immunologically heterogeneous. We demonstrated this recently through the analysis of myeloid cells purified from 20 human GBM specimens. The study identified that GBM subtypes with high expression of Pyk2/FAK, EGFR or PDGFRA correlate with distinct cytokines expression profile of tumor infiltrating myeloid cells (Nunez et al., 2021).
Figure 1 (Figure is attached). Heat map of Pearson correlation matrix for microglia cytokine and chemokine gene expression and the corresponding cell-surface receptor gene expression, as well as Pyk2 and FAK gene and protein expression in human glioma cells. Cytokines/chemokines with a strong correlation (0.6–1.0) selected for further study are highlighted in red (n=20).
Additionally, the Hambardzumyan group demonstrated in murine models that myeloid cell composition in tumors, generated with different driver mutations, showed marked differences (Chen et al, 2020). They found that Nf1-silensed tumors had the highest level of Iba1 immunopositivity, compared to the PDGFB-overexpressing and EGFRvIII tumors. This same observation held true for their corresponding human GBM subtypes. Additionally, the majority of myeloid cells in PDGFB-overexpressing and EGFRvIII-expressing tumors were monocyte-derived, while in Nf1-silenced murine tumors the majority of the myeloid cells were brain-resident microglia. A similar trend was previously documented in human samples (Gabrusievicz et al, 2016).
For the current study we selected the GL261/C56Bl/6 model due to its broad use, extensive characterization and high reproducibility. Currently, based on literature background and on data presented in this manuscript, we continue the study directed at investigation of dynamic of cytokine expression profile during the post-surgical period in tumors with distinct driver mutations with use of murine immunocompetent model. This investigation is the direct continuation of study presented, and results will be published as an individual report.
We added discussion of immunological heterogeneity of GBMs and relation to tumor genetic profile to the Discussion section (lines 623-633).
Nuñez RE, Del Valle MM, Ortiz K, Almodovar L, Kucheryavykh L. Microglial Cytokines Induce Invasiveness and Proliferation of Human Glioblastoma through Pyk2 and FAK Activation. Cancers (Basel). 2021 Dec 7;13(24):6160. doi: 10.3390/cancers13246160.
Chen Z, Herting CJ, Ross JL, Gabanic B, Puigdelloses Vallcorba M, Szulzewsky F, Wojciechowicz ML, Cimino PJ, Ezhilarasan R, Sulman EP, Ying M, Ma'ayan A, Read RD, Hambardzumyan D. Genetic driver mutations introduced in identical cell-of-origin in murine glioblastoma reveal distinct immune landscapes but similar response to checkpoint blockade. Glia. 2020 Oct;68(10):2148-2166. doi: 10.1002/glia.23883.
Gabrusiewicz K, Rodriguez B, Wei J, Hashimoto Y, Healy LM, Maiti SN, … Heimberger AB (2016). Glioblastoma-infiltrated innate immune cells resemble M0 macrophage phenotype. JCI Insight, 1(2), 85841 10.1172/jci.insight.85841
- I am curious to see how close these mouse models are to actual human surgical tissues. Can the authors show cytokine data for Day 0 after surgical resection in surgically obtained human tissue, or at the least discuss previous studies looking at this issue?
Comparative single-cell profiling of myeloid cells between GL261/C57Bl/6 model and human GBM specimens observed conservation between the human and mouse GBM immune compartment (Pombo Antunes et al, 2021). However, it was indicated that the GL261 immune landscape seemed to more closely resemble human recurrent tumors. We included discussion of differences between our study results and human data in lines 617-622.
Additionally, we included discussion of a recent study, performed on orthotopic xenograft tumor resection model in rats, in lines 614-617. The study indicates up-regulation of VEGF and interleukin signaling in regrown tumors and increased infiltration of myeloid cells into the recurrent tumors, which is consistent with our results, obtained in GL261/Bl/6 model.
We also added patient dataset analysis to the results section (lines 404-426 and Supplementary Fig. 3 and 4) to confirm the relevance of the identified cytokines for overall survival in GBM patients. Additionally, we provided a reference from the multiplex study, that identified significant up-regulation of IL-6, IL-1β, TNF-α, IL-10, VEGF, FGF-2, IL-8, IL-2, and GM-CSF in human GBM specimens, which correlated with tumor grade, proliferation markers, and clinical aggressiveness in glioblastomas (lines 538-540).
- It would be great if the authors could provide a figure illustrating the dynamic changes in cytokine expression at the different time points after surgery.
We added a Fig. 6, schematically illustrating the dynamic changes of cytokine expression in postsurgical period.
Minor points:
- Nation Cancer Institute (line 99) should be National Cancer Institute.
This typo has been fixed.

Round 2
Reviewer 2 Report
If GBMs are immunologically heterogeneous, then that is further reason that you cannot jump to conclusions from just one model (cell line). Why are you trying to divide your data into multiple papers, when one big paper would be much more convincing.
At the least, you need to state in the discussion that using only one model is a major limitation of this study.
Author Response
If GBMs are immunologically heterogeneous, then that is further reason that you cannot jump to conclusions from just one model (cell line). Why are you trying to divide your data into multiple papers, when one big paper would be much more convincing.
In conclusion we state that our findings are related to GL261/C57Bl/6 glioma implantation model (line 639)
At the least, you need to state in the discussion that using only one model is a major limitation of this study.
We added the requested statement in line 634 and outlined the avenue for the future study development.
